# OBJECTIVE MISMATCH IN MODEL-BASED REINFORCEMENT LEARNING

## ABSTRACT

Model-based reinforcement learning (MBRL) has been shown to be a powerful framework for data-efficiently learning control of continuous tasks. Recent work in MBRL has mostly focused on using more advanced function approximators and planning schemes, leaving the general framework virtually unchanged since its conception. In this paper, we identify a fundamental issue of the standard MBRL framework – what we call the *objective mismatch issue*. Objective mismatch arises when one objective is optimized in the hope that a second, often uncorrelated, metric will also be optimized. In the context of MBRL, we characterize the objective mismatch between training the forward dynamics model w.r.t. the likelihood of the one-step ahead prediction, and the overall goal of improving performance on a downstream control task. For example, this issue can emerge with the realization that dynamics models effective for a specific task do not necessarily need to be globally accurate, and vice versa globally accurate models might not be sufficiently accurate locally to obtain good control performance on a specific task. In our experiments, we study this objective mismatch issue and demonstrate that the likelihood of the one-step ahead prediction is not always correlated with downstream control performance. This observation highlights a critical flaw in the current MBRL framework which will require further research to be fully understood and addressed. We propose an initial method to mitigate the mismatch issue by re-weighting dynamics model training. Building on it, we conclude with a discussion about other potential directions of future research for addressing this issue.

## 1 INTRODUCTION

Model-based reinforcement learning (MBRL) is a popular approach for learning to control nonlinear systems that cannot be expressed analytically (Bertsekas, 1995; Sutton and Barto, 2018; Deisenroth and Rasmussen, 2011; Williams et al., 2017). MBRL techniques achieve the state of the art performance for continuous-control problems with access to a limited number of trials (*e.g.* Chua et al. (2018); Wang and Ba (2019)) and in controlling systems given only visual observations with no observations of the original system's state (*e.g.* Hafner et al. (2018); Zhang et al. (2018)). MBRL approaches typically learn a *forward dynamics model* that predicts how the dynamical system will evolve when a set of control signals are applied. This model is classically fit with respect to the maximum likelihood of a set of trajectories collected on the real system, and then used as part of a control algorithm to be executed on the system (e.g., model-predictive control).

In this paper, we highlight a fundamental problem in the standard MBRL learning scheme: the *objective mismatch issue*. The learning of the forward dynamics model is decoupled from the subsequent controller and policy that it induces through the optimization of two different objective functions – negative log-likelihood (or its equivalent for deterministic models, *e.g.*, the RMSE) of the single- or multi-step look-ahead prediction for the dynamics model, and task performance (*i.e.*, reward) for the policy optimization. While the use of negative log-likelihood (NLL) for system identification is an historically accepted objective, it results in optimizing an objective that does not necessarily correlate to the actual performance. The contributions of this paper are to: 1) identify and formalize the problem of objective mismatch in MBRL; 2) examine the signs of and the effects of objective mismatch on simulated control tasks; 3) propose a initial mechanism to mitigate objective mismatch; 4) discuss the impact of objective mismatch on existing MBRL and outline future directions to address this problem.

Dynamics $f_\theta$ — Control → Policy $\pi_\theta(x)$ — Interacts → Environment

**Training:** Maximum Likelihood  **Objective Mismatch**  Responses

Trajectories  State Transitions  Reward

Figure 1: Objective mismatch in MBRL arises when a dynamics model is trained to maximize the likelihood but then used for the policy to maximize a reward signal that is not used during training.

## 2 MODEL-BASED REINFORCEMENT LEARNING

We now outline the MBRL formulation used in the paper. At time $t$, we denote the state $s_t \in \mathbb{R}^{d_s}$, the actions $a_t \in \mathbb{R}^{d_a}$, and the reward $r(s_t, a_t)$. We say that the MBRL agent acts in an environment governed by a state transition distribution $p(s_{t+1}|s_t, a_t)$. We denote a parametric model to approximate this distribution with $p_\theta(s_{t+1}|s_t, a_t)$. MBRL follows the general approach of an agent acting in its environment, learning a model of said environment, and then

---

**Algorithm 1:** Model-based RL

**Data:** Initialize $\mathcal{D}$ from random actions
**while** *Improving* **do**
    Train model $p_\theta(s_{t+1}|s_t, a_t)$ on $\mathcal{D}$
    Collect data $\mathcal{D}'$ with controller using $p_\theta$
     in real environment
    Aggregate dataset $\mathcal{D} = \mathcal{D} \cup \mathcal{D}'$

---

leveraging the model to act more effectively, summarized in Alg. 1. While iterating over parametric control policies, the agent collects measurements of state, action, next-state and forms a dataset $\mathcal{D} = \{(s_n, a_n, s'_n)\}_{n=1}^N$, where $N$ is the sum of the length of all episodes. With the dynamics data $\mathcal{D}$, the agent learns the environment in the form of a neural network forward dynamics model, learning an approximate dynamics $p_\theta$. This dynamics model is leveraged by a controller that takes in the current state $s_t$ and returns an action sequence $a_{t:t+T}$ maximizing the expected reward $\mathbb{E}_{\pi_\theta(s_t)} \sum_{i=t}^{t+T} r(s_i, a_i)$, where $T$ is the predictive horizon and $\pi_\theta(s_t)$ is the set of state transitions induced by the model $p_\theta$. In our paper, we primarily use probabilistic networks designed to minimize the NLL of the predicted parametric distribution $p_\theta$, denoted as $P$, or ensembles of probabilistic networks denoted $PE$, and compare to deterministic networks minimizing the mean squared error (MSE), denoted $D$ or $DE$. Unless otherwise stated we use the models as in PETS (Chua et al., 2018) with an expectation-based trajectory planner and a cross-entropy-method (CEM) optimizer.

## 3 OBJECTIVE MISMATCH AND ITS CONSEQUENCES

**The Origin of Objective Mismatch: The Subtle Differences between MBRL and System Identification** Many ideas and concepts in model-based RL are rooted in the field of optimal control and system identification (Bertsekas, 1995; Zhou et al., 1996; Kirk, 2012; Bryson, 2018; Sutton and Barto, 2018). In system identification, the main idea is to use a two-step process where we first generate on the robot (optimal) elicitation trajectories $\tau$ to fit a dynamics model (typically analytical), and subsequently we apply this model to a specific task. This particular scheme has multiple assumptions: 1) the data collected cover the entire state-action space (done by appropriate elicitation trajectories); 2) the presence of large (virtually infinite) amount of data; 3) the global nature of the model resulting from the system identification process (which should be generalizable to a large range of tasks). With these assumptions, the general idea of system identification is effectively to collect large amount of data covering the whole state-space to create

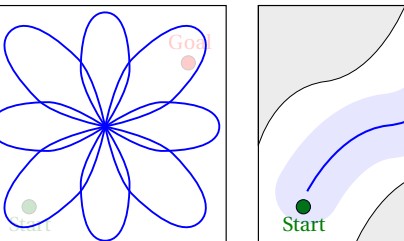

Figure 2: Sketch of the state-action space for system identification and MBRL. (*Left*) In system identification, the elicitation trajectories are designed off-line to cover the entire state-action space, without considering a specific task. (*Right*) In MBRL instead, the data collected during the learning is often concentrated in trajectories towards the goal, with other parts of the state-action space being completely unexplored (grey area).

once a global model that is sufficiently accurate that we can at deployment time specify any desired task, and still obtain good performance.

When adopting the idea of learning the dynamics model used in optimal control for MBRL, it is important to consider if these assumptions still hold. The assumption of virtually infinite data is visibly in tension with the explicit goal of MBRL which is to reduce the number of interactions with the environment by being "smart" about the sampling of new trajectories. In fact, in MBRL the offline data collection performed via elicitation trajectories is largely replaced by on-policy sampling (with the exception of some initial motor babbling (Chua et al., 2018)) in order to explicitly reduce the need to collect large amount of data. Moreover – in practice – in the MBRL setting the data will not usually cover the entire state-action space, since they are generated by a policy trying to optimize one specific task. In conjunction with the use of non-parametric models (i.e., not analytical model), this results in learned models that are strongly biased towards capturing the distribution of the data, and that are only locally accurate. Nonetheless, this is at first sight not an issue since the common MBRL setting consider the learning aimed at solving only one specific task, and rarely test for generalization capabilities of the learned dynamics.

In practice, we can now see how the assumptions and goals of system identification are in contrast with the ones of MBRL. Understanding these differences and the downstream effects on algorithmic approach is crucial to design new families of MBRL algorithms.

**Objective Mismatch**  During the MBRL process of iteratively learning a controller, the reward signal from the environment is diluted by the training of a forward dynamics model with a independent metric, as showing in Fig. 1. In our experiments, we highlight that the minimization of some network training cost does not hold a strong correlation to maximization of episode reward. As dynamic environments becoming increasingly complex in dimensionality, the assumptions of collected data distributions become weaker and over-fitting to different data poses an increased risk.

Formally, the problem of objective mismatch appears as two de-coupled optimization problems repeated over many cycles of Alg. 1, shown in Eq. (1a,b), which could be at the cost of minimizing the final reward. This loop becomes increasingly difficult to analyze as the dataset used for model training changes with each experimental trial – a step that is needed to include new data from previously unexplored states. In this paper we characterize the problems introduced by the interaction of these two optimization problems, but avoid to consider the interactions added by the changes in data distribution during the learning process, as this would significantly increase the complexity of the analysis. In addition, we discuss potential solutions, but do not make claims about the best way to do so, which is left for future work.

$$\textbf{Training: } \arg\min_\theta \sum_{i=1}^{N} \log p_\theta(s_i'|s_i, a_i), \quad \textbf{Control: } \arg\max_{a_{t:t+T}} \mathbb{E}_{\pi_\theta(s_t)} \sum_{i=t}^{t+T} r(s_i, a_i) \quad \text{(1a,b)}$$

## 4 IDENTIFYING OBJECTIVE MISMATCH

In this section, we experimentally study the issue of objective mismatch in MBRL to answer the following questions: 1) Does the distribution of models obtained from running a MBRL algorithm show a strong correlation between NLL and reward? 2) Are there signs of sub-optimality in the dynamics models training process that could be limiting performance? 3) What model differences are reflected in reward but not in NLL?

**Experimental Setting**  In our experiments, we use two popular RL benchmark tasks: the cartpole and half cheetah. For more details on these tasks see Chua et al. (2018). For our cartpole experiments, we aggregate dynamics models at each episode from 50 runs of PETS (20 trials per run) paired with the on-policy episode reward achieved, giving the number of models $M_{cp} = 1000$. Our half cheetah experiments consist of 8 runs (300 trials per run) of PETS, tabulating the reward with the dynamics models again, yielding $M_{hc} = 2400$. All dynamics models are of depth 3, hidden width 500, and use the $\tanh$ activation function. With a large set of dynamics models in the feasible space of on-policy models, we then used a series of control datasets to evaluate the NLL of the dynamics models versus performance. The datasets we use are designed to investigate how different assumptions made in MBRL trickle down into final performance. We start with expert datasets to

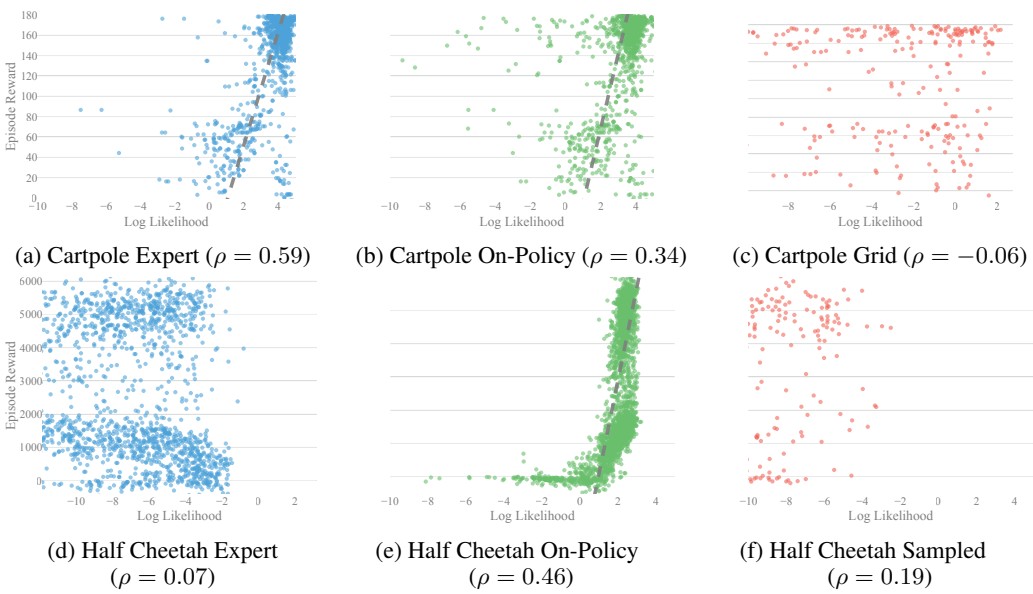

Figure 3: The distribution of dynamics models from our PETS experiments plotting in the LL-Reward space on three datasets, with correlation coefficients $\rho$. (*Notes: we are using log likelihood (LL) here rather than NLL for easier interpretability and the reward at each point is the mean of 10 trials with the CEM optimizer, to disentangle the stochasticity of MPC.*) The numbers of models evaluated are $M_{cp} = 1000$ and $M_{hc} = 2400$. For the datasets shown in (c,d,f) many of the LL's are extremely high and outside of the range of the figure, but the trend lines and correlation coefficients are calculated on all points. There is a trend of high reward to 'good' LL that breaks down as the datasets contain more of the state-space than only expert trajectories. Trend lines are regressed from total least squares, and included when the sum of squares solution implemented converges.

test if on-policy performance is only linked to having adequately explored the environment. As a baseline, we compare the expert data to datasets collected by standard data aggregation procedures when learning. Finally, a dataset representing the entire state space is generated as a grid over the state-space or by sampling from reasonable values and measuring the transitions in order to test how complete dynamics models relates to model accuracy and final performance. Specifically, the cartpole datasets are as follows: an expert selection of episodes data with $r > 179$ (2, 400 points), a standard on-policy dataset aggregated from the end of a PETS run (3, 780 points), and a grid of datapoints to approximate the full system dynamics (16, 807 points). For half cheetah we compared three similar datasets: an expert dataset from a PETS run of half cheetah using the true dynamics for planning (3, 000 points), an on-policy dataset aggregated from PETS (90, 900 points), and an uniform sampled dataset attempting to approximate the full system dynamics (200, 000 points).

## 4.1 EXPLORATION OF MODEL LOSS VS EPISODE REWARD SPACE

In the standard MBRL framework, it is assumed that there is a clear proportional relationship between model testing loss and on-policy performance. We here show that this is a strong assumption even in simple domains, and on specific on-policy data. The relationships between model accuracy on data representing the full environment space and reward show no concrete trend in Fig. 3c,f. The simplicity of the cartpole environment results in quick learning and a concentration of networks around peak reward. The distribution of rewards versus log-likelihood (LL) is shown in Fig. 3a-c, where there is a trend of increased reward as LL decreases, but there is substantial variance and points of disagreement. This bi-model distribution on the half cheetah expert data set, shown in most clearly Fig. 3d, relates to a failure mode in early half cheetah trials where the agent enters a unrecoverable state. The contrast between Fig. 3e and Fig. 3d,f shows a substantial difference in the transitions represented within the datasets. The multiple modes by which half cheetah runs during different stages of learning warrants more investigation into how new data should be incorporated into a training dataset while maintaining the knowledge of older data.

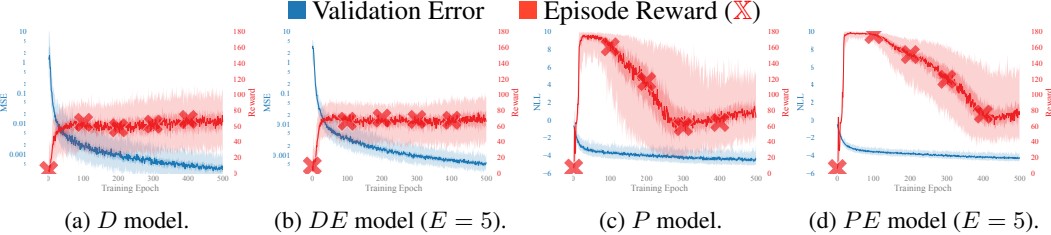

(a) $D$ model.   (b) $DE$ model ($E = 5$).   (c) $P$ model.   (d) $PE$ model ($E = 5$).

Figure 4: The reward versus epoch when re-evaluating the controller leveraging a dynamics model at each training epoch for different types of dynamics models. Even for the simple cartpole environment, networks of width 500 and depth 3 cannot learn the entire grid dataset: simple models ($D$, $DE$) fail to achieve full performance, while more advanced models reach higher performance but eventually over-fit to available data ($P$, $PE$). The over-fitting of the $P$ model is further evaluated in Fig. 5a.

These results show that objective mismatch is not preventing MBRL from functioning, which would appear as model loss being fully dissociated with performance and the plots shown lacking any coherent shape. Rather, the mismatch we show likely is a ceiling on the performance of current MBRL algorithms by reducing the correlation between model accuracy and evaluation reward. The distributions in this section of on-policy data in Fig. 3b,e show a *noisy trend* of higher reward with better model loss and not the *guarantee* of improvement that is expected when training a better model.

## 4.2 MODEL LOSS VS EPISODE REWARD DURING TRAINING

This section explores the relationship between how model training impacts performance at the per-epoch level. These experiments also shed light onto the impact of the strong dataset assumptions outlined in Sec. 3. Figures 4 and 5 are generated by re-evaluating the controller leveraging a dynamics model at each training epoch. As a dynamics model is trained, there are two key inflection points - the first is the training epoch where episode reward is maximized, and the second is when error on the validation set is minimized. When training on data from the entire state-space, our experiments show that the reward is maximized at a drastically different time than when validation loss is minimized. These experiments are focused on showing the disconnect between three standard practices in MBRL a) the assumption that the dynamics data can express large portions of the state-space when collected on policy, b) the idea that simple neural networks can satisfactorily capture complex dynamics, c) and the practice that model training is a simple optimization problem disconnected from reward. For the cartpole environment with the grid dataset, Fig. 4 shows that the performance of the controller degrades for $P$, $PE$ models and fails to adequately learn with the $D$, $DE$ models. For cartpole, training on on-policy data replicates results of the trials ($> 175$ reward), shown in Fig. 5b while the expert dataset fails to achieve any learned performance ($< 20$ reward) because it lacks a robust enough dataset for stable prediction generation during stochastic action sampling. Fig. 5 highlights how the trained models are able to represent other datasets that they are trained on,

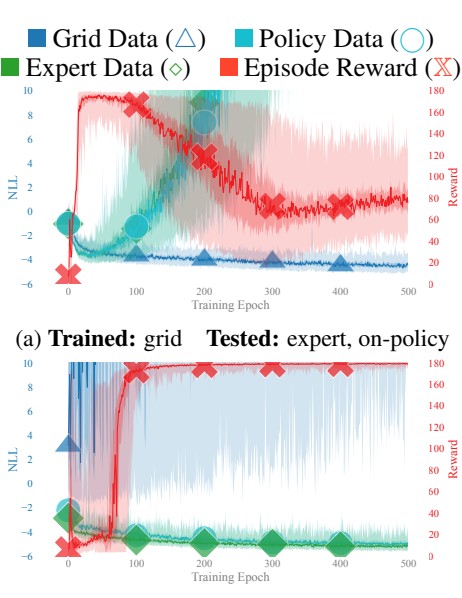

(a) **Trained: grid   Tested: expert, on-policy**

(b) **Trained: on-policy   Tested: expert, grid**

Figure 5: The effect of the dataset choice on model ($P$) training and accuracy in different regions of the state-space. (*Top*) when training on the complete dataset, the model begin over-fitting to the on-policy data even before the performance drops in the controller. (*Bottom*) A model trained only on policy data does not accurately model the entire state-space.

which points to the need for fine-tuning of that assumption on datasets made at the outset between the state space and that collected on-policy. There is no information indicating the on-policy dataset will lead to a complete dynamics understanding because on-policy and grid show little correlation in evaluation loss. When training the gird on cartpole, the fact that the on-policy data diverges before

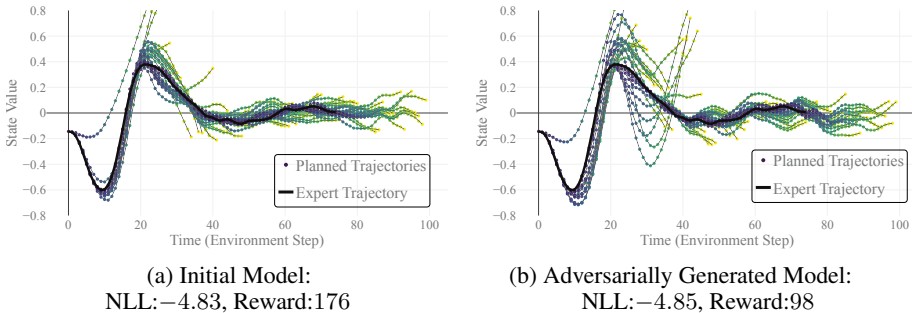

(a) Initial Model:
NLL:$-4.83$, Reward:$176$

(b) Adversarially Generated Model:
NLL:$-4.85$, Reward:$98$

Figure 6: Planned trajectories along the expert trajectory for the initial model and the adversarially generated model trained to lower the reward. It can be seen how the planned trajectories are qualitatively similar except for the peak at $t = 25$. There, the adversarially generated model learned that applying a small nudge to the dynamics model at the right place/moment yield to significantly influencing the control outcome with minimal change in terms of NLL.

the reward decreasing is encouraging as objective mismatch may be preventable in simple tasks, but the problem becomes increasingly complex as the dynamics data distribution becomes large and multi-modal, as in half cheetah. Similar experiments on half cheetah data failed to achieve a matching reward to the full trials because the training process in PETS involves incremental network with dataset aggregation. When training on a single set of dynamics data for many epochs the controller did not achieve comparable reward to networks incrementally trained on new data after each episode.

### 4.3 DECOUPLING MODEL LOSS FROM CONTROLLER PERFORMANCE

When two dynamics models evaluated on a control dataset have similar validation loss, there can be a large variance in the reflected control policy. In this section, we explore how differences in dynamics models are qualitatively reflected in control policies to show the reader that a accurate dynamics model does not guarantee performance.

**Adversarial attack on model performance** We performed an adversarial attack (Goodfellow et al., 2015; Szegedy et al., 2013) on a neural network dynamics model so that it attains a good likelihood but poor reward to continue illustrating objective mismatch. We start with a dynamics model that achieves high likelihood and high reward and tweak the parameters so that it continues achieving high likelihood but has a low reward. We fine-tune the network's last layer to minimize the episode reward, with a large penalty if the model validation likelihood drops below the original value. We use a zeroth-order optimizer (CMA-ES) because the reward is generally non-differentiable. As a starting point for this experiment we sampled a $P$ dynamics model from the last trial of a PETS run

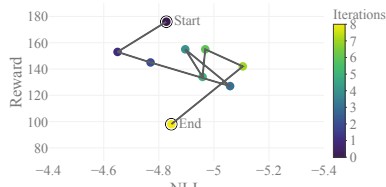

Figure 7: Convergence of the CMA-ES population's best member over iterations to the minimum reward achieved with comparable NLL.

on cartpole. This model achieves reward of $176$ and has a NLL of $-4.827$ on it's on-policy training dataset. Using CMA-ES, we reduced the on-policy reward of the model to $98$ – a $45\%$ decrease in performance compared to the original model – while slightly improving the NLL to $-4.845$; the CMA-ES convergence over population iterations is shown in Fig. 7 and the difference between the two models is visualized in Fig. 6. Fine tuning of all model parameters would be even more likely to find sub-optimal performing controllers with low model loss because the output layer consists of about $1\%$ of the total model parameters. This experiment shows that the model parameters that achieve a low model loss inhabit a broader space than the subset that also achieves high reward.

**Sampling models with similar NLLs, different rewards** To better understand the objective mismatch, we also compared how a difference of model loss can impact a control policy. We sampled models with similar NLL's and extremely different rewards from Fig. 3d-e and visualized the chosen optimal action sequences along an expert trajectory. The control policies and dynamics models appear to be converging to different regions of state spaces. In these visualizations, there is not a emphatic

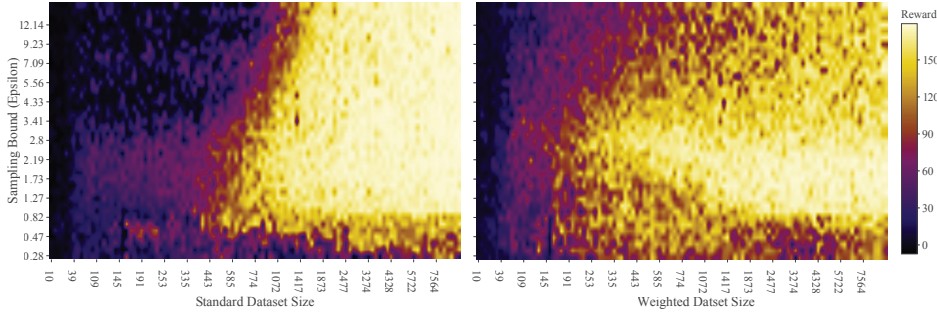

Figure 8: Mean reward of PETS trials, with and without model re-weighting, on a log-grid of dynamics model training sets with number of points S $\in [10, 10000]$ and sampling expert-distance bounds $\epsilon \in [.28, 15.66]$. The re-weighting shows an ability to learn moderate performance at substantially lower number of datapoints, but suffers from increased variance in larger set sizes. The performance of PETS declines when the dynamics model is trained on points too near to the expert dataset because the model lacks robustness when running online with the stochastic MPC.

reason why the models achieved different reward, so further study is needed to quantify the impact of model differences. The interpretability of the difference between models and controllers will be important to solving the objective-mismatch issue.

## 5 ADDRESSING OBJECTIVE MISMATCH DURING MODEL TRAINING

Tweaking dynamics model training can partially mitigate the problem of objective mismatch. While keeping the NLL minimization standard in MBRL, the trainer can prioritize state transitions associated with an expert trajectory over tuples further from expert, and dynamics model will use its capacity to learn relevant tasks more quickly. We show improved sample efficiency on Cartpole via re-weighting network training by a measure of Euclidean distance in the state-action space.

Given a element of a state space $(s_i, a_i)$, we can quantify the distance of any two tuples $\{(s_i, a_i, s'_i), i = 1, 2\}$ as $d = ||[s_1, a_1] - [s_2, a_2]||$. With this distance, we re-weight the loss, $l(y)$, of points further from an expert policy to be lower, so that points in the expert trajectory get a weight $\omega(y) = 1$, and points at the edge of the grid dataset used in Sec. 4 get a weight $\omega(y) = 0$ (in our experiments, multiple monotonic functions from 0 to 1 had similar results). With this notion of distance and weighting, we used the expert dataset discussed in Sec. 4 as a distance baseline. We generated a base dataset consisting of $25,000,000$ tuples of $(s, a, s')$ by uniformly sampling across the state and action space of cartpole. We taxonomized this dataset by taking the minimum orthogonal distance, $d^*$, from each of the points to the 200 element dataset from one expert trajectory that achieved a reward of $180$. To create different datasets that range from near-expert to nearly uniform across the state space, we vary the distance bound, $\epsilon$, and number of points, S, trained on. For each sampling bound, $\epsilon$, we sample 5 different datasets such that $d_i^* < \epsilon$ from our distance-tabulated random dataset, and for each dataset $(S, \epsilon)$ we trained 5 $P$ models, giving 25 episodes from PETS algorithm on the cartpole task for each point in the heatmap shown in Fig. 8. This simple form of re-weighting the neural network loss, shown in Eq. (2a,b,c), demonstrated an improvement in sample efficiency to learn the cartpole task, as seen in the left halves of each plot in Fig. 8. Developing an iterative method to re-weight samples in an online training method has the potential to further improve the sample efficiency of MBRL baselines.

$$\textbf{Weighting} \ \ \omega(y) = \frac{e^{d^*(y)}}{e} \qquad \textbf{Standard} \ \ l(\hat{y}, y) \qquad \textbf{Re-weight} \ \ l(\hat{y}, y) \cdot \omega(y) \qquad \text{(2a,b,c)}$$

## 6 DISCUSSION, RELATED WORK, AND FUTURE WORK

*Objective mismatch* seriously impacts the performance of MBRL. Our experiments have gone deeper into this fragility in the context of the state-of-the-art MBRL algorithms from an analysis perspective rather than from a solution perspective. Beyond the re-weighting of the NLL presented in Sec. 5, here we summarize and discuss the key pieces in the community starting to address this issue.

**Learning the dynamics model to optimize the task performance** Most relevant to our work are research directions on controllers that directly connect the reward signal back to terms of the controller. In theory, this exactly solves the model mismatch problem we discuss in this paper but in practice the current approaches are borderline intractable and have proven difficult to scale to complex systems. One popular way to do this is by designing systems that are fully differentiable so that the task reward can be backpropagated all the way to the dynamics. This has been investigated with differentiable MPC (Amos et al., 2018), Path Integral control (Okada et al., 2017; Pereira et al., 2018), and stochastic optimization (Donti et al., 2017). Universal Planning Networks (Srinivas et al., 2018) propose a differentiable planner that unrolls gradient descent steps over the action space of a planning network. An alternative explored in Bansal et al. (2017) is to use a zero-order optimizer instead (Bayesian optimization specifically) which can be used to learn locally linear dynamics to maximize the controller's performance without having to compute gradients explicitly.

**Shaping the cost or reward** Closely related are methods that, instead of learning a dynamics model optimizing task performance, fix the dynamics model and fine-tune the cost function of a controller to optimize, *e.g.*, imitation loss Trimpe et al. (2014); Talvitie (2018); Tamar et al. (2017).

**Add heuristics to the dynamics model structure or training process to make control easier** If it is infeasible or intractable to shape the cost or dynamics components of a controller, adding heuristics to the structure or training process of the dynamics model is reasonable and can give state-of-the-art results in many settings. One challenge in these heuristics is that they may be unstable and difficult to fix or improve when they do not work in new environments. These heuristics can manifest in the form of learning a latent space that is locally linear, *e.g.*, in Embed to Control and related methods (Watter et al., 2015; Banijamali et al., 2017), by enforcing that the model makes long-horizon predictions (Ke et al., 2019), ignoring parts of the state space that actions can not control (Ghosh et al., 2018), detecting and correcting when a predictive model steps off the manifold of reasonable states (Talvitie, 2017), adding reward signal prediction on top of the latent space Gelada et al. (2019), adding noise when training transitions Mankowitz et al. (2019). among other approaches Jonschkowski et al. (2018); Ke et al. (2019); Miladinović et al. (2019); Ichter and Pavone (2019); Singh et al. (2019).

**Using incorrect models** Instead of fixing the issues directly in the model or cost parameters, one can assume that the approximations are always incorrect but to use them anyway and potentially try to correct them Abbeel et al. (2006); Sorg et al. (2010); Joseph et al. (2013); Nagabandi et al. (2018).

**Add inductive biases to the controller** Prior knowledge and inductive biases can be added to the controller in the form of hyper-parameters such as the horizon length, or by penalizing unreasonable control sequences by using, *e.g.*, a slew rate penalty. These heuristics can significantly improve the performance if done correctly but can be difficult to tune. Jiang et al. (2015) use complexity theory to justify using a *short* planning horizon with an approximate model to reduce the the class of induced policies. Whitney and Fergus (2018) empirically show that long horizons are better for some tasks.

**Continuing Experiments** Our experiments represent an initial exploration into the challenges of objective mismatch in MBRL, but more analysis is needed. Our exploration in Sec. 4.2 is limited to cartpole due to computational challenges of training with large dynamics datasets and Sec. 4.3 could be strengthened by defining quantitative comparisons in controller performance other than episode reward. Additionally, the effects of model mismatch should be quantified in other state of the art algorithms in MBRL such as MBPO (Janner et al., 2019) and POPLIN (Wang and Ba, 2019).

## 7 CONCLUSION

This paper identifies, formalizes and analyzes the issue of objective mismatch in MBRL. This fundamental disconnect in objectives between the training of the model w.r.t. the likelihood, and the overall task reward emerges from the subtle differences at the origins of MBRL with assumptions that have not been fully adapted. Experimental results highlighted the negative effects that objective mismatch have on the performance of a current state of the art MBRL algorithm. Our results are an exposition of current limitations and an encouragement to investigate nuance in current practices. In providing a first insight on the issue of objective mismatch in MBRL, we hope future work will more deeply examine these underlying assumptions. With our re-weighting method to mitigate mismatch, which also shows improvements in sample efficiency, we share our more thoughts about promising directions of future research to address this issue. We believe that fundamentally understanding and addressing the objective mismatch issue will contribute to improving the performance of MBRL.

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

APPENDIX

## A    EFFECT OF DATASET DISTRIBUTION WHEN LEARNING

Learning speed can be slowed by many factors in dataset distribution, such as adding additional irrelevant transitions. When extra transitions from a specific area of the state space are included in the training set, the dynamics model will spend increased expression on these transitions. NLL of the model will be biased down as it learns this data, but it will reduce the learning speed as new, more relevant transitions are added to the training set.

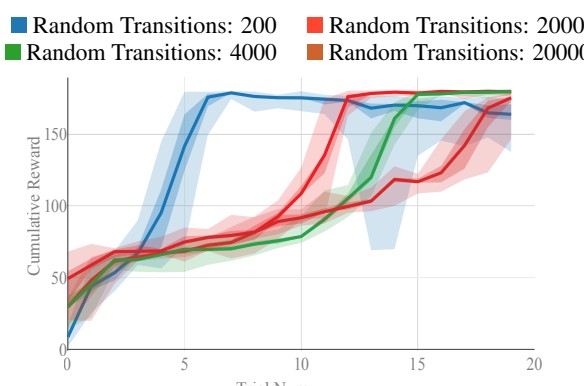

Running cartpole random data collection with a short horizon of 10 steps (while forcing initial babbling state to always be 0), for 20, 200,400 and 2000 babbling rollouts (that sums up to 200, 2000, 4000 and 20000 transitions in the dataset finally shows some regression in the learning speed for runs with more useless data in the motor babbling. This data highlights the importance of careful exploration vs exploitation tradeoffs, or changing how models are trained to be selective with data.

Figure 9: Cartpole (Mujoco simulations) learning efficiency is suppressed when additional data not relevant to the task is added to the dynamics model training set. This effect is related to the issue of objective mismatch because model training needs to account for potential off-task data.

## B    TASK GENERALIZATION IN SIMPLE ENVIRONMENTS

In this section, we compare the performance of a model trained on data for the standard cartpole task (x position goal at 0) to policies attempting to move the cart to different positions in the x-axis. Fig. 10 is a learning curve of PETS with a PE model using the CEM optimizer. Even though performance levels out, the NLL continues to decrease as the dynamics models accrue more data. With more complicated systems, such as halfcheetah, the reward of different tasks verses global likelihood of the model would likely be more interesting (especially with incremental model training) - we will investigate this in future work. Below, we show that the dynamics model generalizes well to tasks close to zero (both positive in (Fig. 11b) and negative positions (Fig. 11a), but performance drops off in areas the training set does not cover as well.

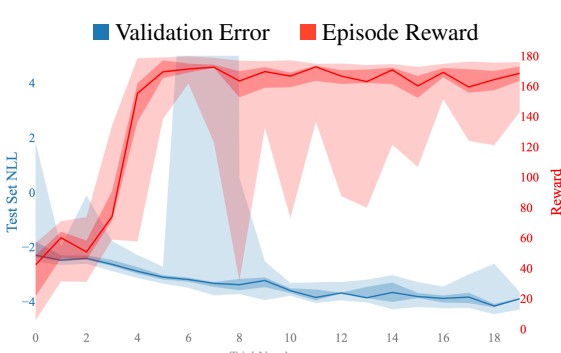

Figure 10: Learning curve for the standard Cartpole task used in this paper ($X_{goal} = 0$). The median reward from 10 trials is plotted with the mean NLL of the dynamics models at each iteration. The reward reaches maximum (180) well before the NLL is at it's minimum.

Below the learning curves in Fig. 12, we include snapshots of the distributions of training data used for these models at different trials, showing how coverage relates to reward in cartpole. It is worth investigating how many points can be removed from the training set while maintaining peak performance on each task.

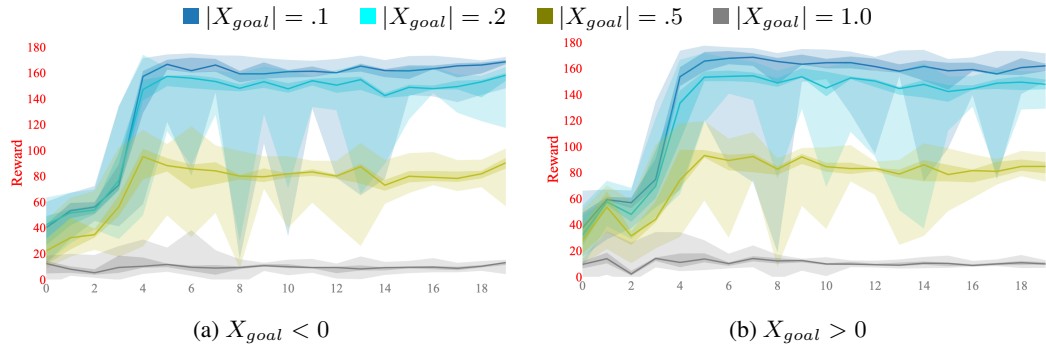

(a) $X_{goal} < 0$        (b) $X_{goal} > 0$

Figure 11: MPC control with different reward functions with the same dynamics models loaded from trials shown in Fig. 10. The cartpole solves tasks further from $0$ proportional to the state space coverage (*Goal further from zero causes reduced performance*). The distribution of $x$ data encountered is shown in Fig. 12.

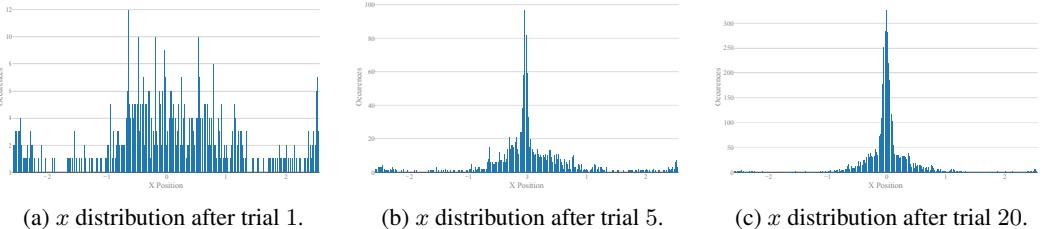

(a) $x$ distribution after trial 1.    (b) $x$ distribution after trial 5.    (c) $x$ distribution after trial 20.

Figure 12: Distribution of $x$ position encountered during the trials shown in Fig. 10. The distribution converges to a high concentration around $0$, making it difficult for MPC to control outside of the area close to $0$.

## C   WAYS MODEL MISMATCH CAN HARM THE PERFORMANCE OF A CONTROLLER

Model mismatch between fitting the likelihood and optimizing the task's reward manifests itself in many ways. Here we highlight two of them and in Sec. 6 we discuss how related work connects in with these issues.

**Long-horizon rollouts of the model may be unstable and inaccurate.**    Time-series or dynamics models that are unrolled for long periods of time easily diverge from the true prediction and can easily step into predicting future states that are not on the manifold of reasonable trajectories. Taking these faulty dynamics models and using them as a smaller part of a controller that optimizes some cost function under a poor approximation to the dynamics. Issues can especially manifest if, *e.g.*, the approximate dynamics do not properly capture stationarity properties necessary for the optimality of the true physical system being modeled.

**Non-convex and non-smooth models may make the control optimization problem challenging** The approximate dynamics might have bad properties that make the control optimization problem much more difficult than on the true system, even when the true optimal action sequence is optimal under the approximate model. This is especially true when using neural network as they introduce non-linearities and non-smoothness that make many classical control approaches difficult.

