# OpenReview forum: "Objective Mismatch in Model-based Reinforcement Learning"
_ICLR.cc/2020/Conference — Reject_

### Official Review · AnonReviewer1 · 2019-10-22
**Official Blind Review #1**

**Rating:** 3

**Review:**

The paper "OBJECTIVE MISMATCH IN MODEL-BASED REINFORCEMENT LEARNING" explores the relationships between model optimization and control improvement in model-based reinforcement learning. While it is an interesting problem, the paper fails at demonstrating really useful effects, and the writting needs to be greatly improved to help reader to focus on salient points.

From my point of view, the main problem of this paper is that it is too messy and it is very difficult to understand what authors want to show, as i) there is a very important lack of experimental details (e.g., main aspects of models and controllers should be clearly stated) and ii) analysis is to wordy, authors should emphasize the message in each part. From the experiments in 4.1, the only thing that I got is from the last sentence "noisy trend of higher reward with better model loss". are these results from LL computed on a validation set ? If not, this is not reallly meaningfull since high LL may only indicate overfitting. If yes, how was the validation data collected ? If the collection is not inline with training it is difficult to understand what we observe since we only need LL to be good on the path from the current to the opitmal policy, not everywhere. Even if the validation data is inline with training, there remains the difficulty of over-fitting in the policy area (for the on-policy experiments at least). Is there something else ?  From 4.2 we observe that it is unsurprisingly better to learn the model from the policy trajectories. From 4.3, we observe that an adversarial is able to reduce rewards without losing in LL. Ok, the adversarial is able to lock the controler in a sub-optimal area while still being good to model the dynamics elsewhere, but what does it show ?  Finally, proposal to cope with the identified mismatch are not clearly explained and not very convincing. Is re-weighting helping in collecting higher rewards ?

From my point of view, this work is in a too preliminary state to be published at ICLR

**Experience Assessment:**

I have read many papers in this area.

**Review Assessment: Checking Correctness Of Derivations And Theory:**

I did not assess the derivations or theory.

**Review Assessment: Checking Correctness Of Experiments:**

I assessed the sensibility of the experiments.

**Review Assessment: Thoroughness In Paper Reading:**

I made a quick assessment of this paper.

---

> ### Author Response · Authors · 2019-11-13
> **Reviewer 1 Author Response**
>
> >> Thank you for your comments. As with all the review, we addressed all of the comments individually, please respond if you would like further clarification.  We hope to improve the quality of the final paper.
>
> R1: From my point of view, the main problem of this paper is that it is too messy and it is very difficult to understand what authors want to show,
>
> >> If you have more specific sentences you would like improved, we are happy to do so.
>
> R1:  as i) there is a very important lack of experimental details (e.g., main aspects of models and controllers should be clearly stated)
> >> The last sentence of our section on model-based reinforcement learning states “Unless otherwise stated we use the models as in PETS (Chua et al., 2018) with an expectation-based trajectory planner and a cross-entropy-method (CEM) optimizer.” We can add the specific numerical details to the appendix. They closely mirror the original implementation used in Chua at al., so they were omitted for space.
>
> R1: and ii) analysis is to wordy, authors should emphasize the message in each part.
>
> >> Again, we did our best to summarize the objectives of each section going in and highlight the conclusion at the end. Let us know where it could be improved.
>
> R1: From the experiments in 4.1, the only thing that I got is from the last sentence "noisy trend of higher reward with better model loss". are these results from LL computed on a validation set ? If not, this is not reallly meaningfull since high LL may only indicate overfitting. If yes, how was the validation data collected ?
>
> >> The section 4.1 specifically details what dataset is used to calculate each log-likelihood. The LL is evaluated on data not used when training, so yes it is a form of a validation set. Please re-read 4.1 for a more specific answer, but we took a very large variety of dynamics models trained when running the PETS algorithm and evaluated the loss on three different datasets (this datasets are all different from the data specifically trained on). The expert dataset is transitions from either an omniscient controller solving the task (cheetah) or a converged solution (cartpole). Then the on-pricy data is a selection of data from one training run to mimic the on-policy distribution, but it is not the same data used when training the models. The grid and sampled datasets are meant to represent the entire state space.
>
> R1: If the collection is not inline with training it is difficult to understand what we observe since we only need LL to be good on the path from the current to the opitmal policy, not everywhere.
>
> >> Our experiments show that if the data trained on is too close to the optimal trajectory, then the controller will have very poor performance. This is due to the stochastic nature of the controller, where it will end up taking some slight perturbations from optimal control, so if the dynamics model does not capture this performance will degrade.
>
> R1: Even if the validation data is inline with training, there remains the difficulty of over-fitting in the policy area (for the on-policy experiments at least). Is there something else ?
>
> >>All of the models are trained in the same manner used in the PETS algorithm, which we showed to match benchmarking performance (please see an expanded comment on this to reviewer #3). If the models are overfitting, it is overfitting in a way that converges to peak performance, which leads us to belief it is a non-issue.
>
> R1:  From 4.2 we observe that it is unsurprisingly better to learn the model from the policy trajectories.
>
> >> This may be unsurprising to one familiar with MBRL research, but many algorithms in MBRL tout the ability to generalize to other tasks then specific task to the policy. This would indicate that when trained on a dense grid of data, the controller should be able to solve a simple problem such as cartpole, but that is not even the case.
>
> (To be continued for character limit)

---

> ### Author Response · Authors · 2019-11-13
> **Reviewer 1 Author Response Continued**
>
> R1: From 4.3, we observe that an adversarial is able to reduce rewards without losing in LL. Ok, the adversarial is able to lock the controler in a sub-optimal area while still being good to model the dynamics elsewhere, but what does it show ?
>
> >> This shows that training a dynamics model to a different local minimum (low NLL) could result in a sub-optimal controller. This effect mirrors the results shown in fig3e where some models with “good” NLL still achieve low reward. When training a model with stochastic methods there is no way to secure coverage of a specific area of the state space, so similar effects could be observed.
>
>
> R1: Finally, proposal to cope with the identified mismatch are not clearly explained and not very convincing. Is re-weighting helping in collecting higher rewards ?
>
> >> Multiple reviewers have brought this point up, and we will expand it’s explanation. Re-weighting is helping to get higher rewards in a particular area of interest – low data  amounts near the expert trajectory. In the regime of dataset size of 100 to 1000 points, the re-weighting is able to solve the task for many different epsilons, but the standard training method only does so with the highest end of the dataset size. This is the region showing the impact of our method, and we included the full picture to show two additional points of interest (best illustrated in the standard training method). 1) data collected too close to expert will not result in a robust controller – seen as the dark area at the bottom of both plots and 2) there is a line where data too far from expert no longer is able to solve the task at a set size – seen as the diagonal line of high vs low reward in the center of the standard plot.

---

### Official Review · AnonReviewer2 · 2019-10-23
**Official Blind Review #2**

**Rating:** 3

**Review:**

The paper claims that it identifies a fundamental issue in model-based reinforcement learning methods. The issue is called objective mismatch, which arises when one objective is optimized (for example, model learning objective) without taking into consideration of another objective (for example policy optimization). The author shows several experiments to illustrate the issue and proposes a method to mitigate it by assigning priorities to samples when training the model.

The issue of objective mismatch is not being noticed the first time. The paper "Reinforcement learning with misspecified model classes" has mentioned similar phenomenon. And other work such as https://arxiv.org/abs/1710.08005 also discussed similar issue.

I disliked the way how the paper is motivated. The paper says “in standard MBRL framework”, what is the standard MBRL framework? I think there is no such standard so far. The claim saying that the mismatch is a crucial flaw in current MBRL framework is too strong. The paper at least completely ignored two broad classes of MBRL methods. The first is value-aware MBRL (which attempts to take into account decision making when learning a model), there are actually many MBRL methods are in this category. Some examples: value prediction network, Predictron, Value-Aware Model Learning. The second class of MBRL approach is Dyna. Several works (continuous deep q-learning with model based acceleration, Organizing Experience: A Deeper Look at Replay Mechanisms for Sample-based Planning in Continuous State Domains, Efficient Model-Based Deep Reinforcement Learning with Variational State Tabulation, Recall Traces: Backtracking Models for Efficient Reinforcement Learning, Hill Climbing on Value Estimates for Search-control in Dyna) show that even with the same model (hence the same model error), using different simulated experiences play a significant role of improving sample efficiency. It is unclear whether model-error plays a decisive role in improving sample efficiency.

The proposed method in section 5 lacks of justification, even a regular ER buffer is asymptotically on-policy, and hence it is indirectly linked with the control performance. It is unclear why introducing the weights based on expert trajectory can be helpful — it can be worse because it should be far away from on-policy distribution.

**Experience Assessment:**

I have published one or two papers in this area.

**Review Assessment: Checking Correctness Of Derivations And Theory:**

I assessed the sensibility of the derivations and theory.

**Review Assessment: Checking Correctness Of Experiments:**

I assessed the sensibility of the experiments.

**Review Assessment: Thoroughness In Paper Reading:**

I made a quick assessment of this paper.

---

> ### Author Response · Authors · 2019-11-13
> **Reviewer 2 Author Response**
>
> >> Thank you for your comments. As with all the review, we addressed all of the comments individually, please respond if you would like further clarification.
>
> R2: The issue of objective mismatch is not being noticed the first time. The paper "Reinforcement learning with misspecified model classes" has mentioned similar phenomenon. And other work such as https://arxiv.org/abs/1710.08005 also discussed similar issue.
>
> >> Thank you for pointing out these papers. If space permits, we will attempt to work them into the related works section. The goal of this paper is to formalize the problem, demonstrate the potential severity of the effects, and call for future research on the topic. We do not claim to be the first to notice this, we are the first to summarize the issue in a complete form so that it is addressed.
>
>
> R2:  I disliked the way how the paper is motivated. The paper says “in standard MBRL framework”, what is the standard MBRL framework? I think there is no such standard so far.
>
> >> Many resources and papers in the area of RL describe the Model-based RL loop as a repetition of collecting data, learning a model, then evaluating a controller, which we are considering. The related work points at new methods where learning a model and a controller simultaneously are considered.
>
> >> Here are some resources that follow our claims, but we can be clearer in the prose and describe this as batch mode model-based RL, or specifically point to Algorithm 1 more frequently: Chua et al. 2018, Berkeley Deep RL course (http://rail.eecs.berkeley.edu/deeprlcourse/), David Silver’s lectures on RL (http://www0.cs.ucl.ac.uk/staff/d.silver/web/Teaching_files/dyna.pdf), In these cases, Dyna is often seen as an intermediate between model-based RL and model-free planning.
>
>
> R2:  The claim saying that the mismatch is a crucial flaw in current MBRL framework is too strong.
>
> >> We will tone this down. The intention is to phrase this as, *it is crucial to understand the underlying mechanism to a process utilized*, rather then, *this is the limiting factor in all MBRL*.
>
> R2:  The paper at least completely ignored two broad classes of MBRL methods. The first is value-aware MBRL (which attempts to take into account decision making when learning a model), there are actually many MBRL methods are in this category. Some examples: value prediction network, Predictron, Value-Aware Model Learning.
>
> >> Thank you for bringing up these papers. There are many papers that have findings relating to this work, and we are doing our best to incorporate as many as possible while stating the position of the problem to model-based RL. The value-aware methods are an important area to be included in the related works. That section can be reworked to include and explain some of the experiments (e.g. Farahmand et al. 2017).
>
> R2:  The second class of MBRL approach is Dyna. Several works ... show that even with the same model (hence the same model error), using different simulated experiences play a significant role of improving sample efficiency. It is unclear whether model-error plays a decisive role in improving sample efficiency.
>
>
>
> >> Dyna could be mentioned as an alternative, but it is a half step removed from model-based RL. It is commonly accepted as a middle ground between a model-free and model-based method. That being said, it does address some of the issues present and could be included.
>
> >> Specifically the point that “using different simulated experiences play a significant role of improving sample efficiency” is important and could be further emphasized in our paper. This should partially be addressed by sampling many random seeds in the paper, but this influence should be acknowledged in the paper as we make many claims where we discuss dataset distribution.
>
>
> R2: The proposed method in section 5 lacks of justification, even a regular ER buffer is asymptotically on-policy, and hence it is indirectly linked with the control performance. It is unclear why introducing the weights based on expert trajectory can be helpful — it can be worse because it should be far away from on-policy distribution.
>
> >> We can edit the text to better motivate the solution. This solution is only for the case where we already have an expert trajectory - wrapping this idea into an online algorithm is a direction for future work. The idea is, if we had an expert trajectory for a given task, we should be able to train a model that focuses on relevant transitions. Adding a higher weight to important transitions hopes to capture this behavior, and not removing other data keeps robustness that is needed with a stochastic controller. The subsection of on-policy  data *near* the expert is the most relevant  for robust control; the data collected in early stages of learning not near the expert transitions will not assist the dynamics model.

---

### Official Review · AnonReviewer3 · 2019-10-24
**Official Blind Review #3**

**Rating:** 3

**Review:**

This paper discusses the old problem of mismatch between the ultimate reward obtained after optimizing a  decision (planning or control) over a probabilistic model (of dynamics) and  the training  objective for the model (log-likelihood). Experiments highlight that the NLL and reward can be very poorly correlated, that improvements in NLL initially improve reward but can later degrade it, and that models with similar NLLs can lead to very different rewards. A  reweighting trick is proposed and summarily evaluated.

I like the topic of this paper but there are several aspects which I see as making it weaker than my acceptance threshold.

First, the paper overclaims in originality. This mismatch problem is not new, it is an instance of a more general issue that end-to-end training and meta-learning try to address, and has been already studied in the context of MBRL by many authors, who actually proposed more substantial solutions. When I read the abstract I had the impression that the paper actually had a theoretical analysis showing the correlation problem, but there is no such thing, only experiments. Section 3 does not actually provide a new insight. Still, the experiments are interesting in that they reveal that the magnitude of the mismatch is probably more serious than most RL researchers believed.

Second, the 'fix' proposed is not well justified nor well tested (e.g. no quantiative comparisons, no comparisons against existing alternative methods to address the same problem, etc). This seriously weakens conclusions like "shows improvements in sample efficiency".

One concern I have about the experiments of fig 3 is that NLL can be really bad, thus distorting rho, which is not a robust measure. So I would only look at NLLs of models with good NLLs, to obtain a more interesting analysis.

Another concern about experiments is that I am not convinced that they were performed with SOTA MBRL methods and hyper-parameters (as demonstrated by SOTA performance on known benchmarks). Otherwise I could easily imagine how the mismatch could be much more severe than in the actual scenarios of interest.

Minor points:


Bottom of page 6 refers to visualizations but I did not see if or where they were shown.

Why the e in the numerator of eq 2e? Seems useless to put any constant there.

The section on 'Shaping the cost or reward' was not clear enough to me (please expand).


**Experience Assessment:**

I have read many papers in this area.

**Review Assessment: Checking Correctness Of Derivations And Theory:**

I carefully checked the derivations and theory.

**Review Assessment: Checking Correctness Of Experiments:**

I carefully checked the experiments.

**Review Assessment: Thoroughness In Paper Reading:**

I read the paper thoroughly.

---

> ### Author Response · Authors · 2019-11-13
> **Reviewer 3 Author Response**
>
> >> Thank you for your comments. As with all the review, we addressed all of the comments individually, please respond if you would like further clarification.  We hope to improve the quality of the final paper.
>
> R3: First, the paper overclaims in originality. This mismatch problem is not new, it is an instance of a more general issue that end-to-end training and meta-learning try to address, and has been already studied in the context of MBRL by many authors, who actually proposed more substantial solutions. When I read the abstract I had the impression that the paper actually had a theoretical analysis showing the correlation problem, but there is no such thing, only experiments. Section 3 does not actually provide a new insight. Still, the experiments are interesting in that they reveal that the magnitude of the mismatch is probably more serious than most RL researchers believed.
>
> >> The goal of this paper is to illustrate the scope of the issue of objective mismatch to encourage future work. Due to the many varieties that this paired optimization problem can take, a general theoretical analysis would prove difficult. The goal of section 3 is to summarize the origins of a problem that is widespread in the area of learning in the specific context of model-based reinforcement learning.
>
>
> R3: Second, the 'fix' proposed is not well justified nor well tested (e.g. no quantiative comparisons, no comparisons against existing alternative methods to address the same problem, etc). This seriously weakens conclusions like "shows improvements in sample efficiency".
>
> >> The fix proposed is an initial attempt to mitigate the problem of objective mismatch, and we hope that better solutions emerge in the future. Due to space limitations, it is also difficult to include more. More thorough methods would be a future paper following this paper proposing the position.
>
> R3: One concern I have about the experiments of fig 3 is that NLL can be really bad, thus distorting rho, which is not a robust measure. So I would only look at NLLs of models with good NLLs, to obtain a more interesting analysis.
>
> >> During these experiments, the large NLLs are already filtered to partially mitigate this. For cartpole, NLL above 20 is filtered, and for half cheetah NLL above 50 is filtered out. We  can reduce the  region of the fit to that of the X-axis shown if it improve understanding.
>
>
> R3: Another concern about experiments is that I am not convinced that they were performed with SOTA MBRL methods and hyper-parameters (as demonstrated by SOTA performance on known benchmarks). Otherwise I could easily imagine how the mismatch could be much more severe than in the actual scenarios of interest.
>
> >> In our experiments, we employ a high-quality re-implementation of PETS which has been thoroughly validated (including against the original code). The difference in performance between our paper, and Chua et al. depends **exclusively** from the use of a more recent version of the MuJoCo simulator. To validate our code, we verified that we can achieve similar performance to Chua et al. when using the older version of MuJoCo, and additionally compared both PETS and SAC with the new MuJoCo simulator to verify that they converge to the same performance (they do).
>
>
> R3: Minor points: Bottom of page 6 refers to visualizations but I did not see if or where they were shown.
>
> >> Good catch. This plots mirrored the results in Figure 6, but needed to be removed due to space limitations. This will be updated in the final version.
>
> R3: Why the e in the numerator of eq 2e? Seems useless to put any constant there.
>
> >> You’re correct, you don’t need the constant and you could change learning rate in training, but we used to constant to not need to change training parameters with and without weighting. We tried a couple other re-weighting techniques and this yielded the best (slightly so) results. The e in the numerator is for normalization. All the re-weighting techniques attempted gave a weight of 1 to points on the expert trajectory, and had some monotonic decrease in weight away from it.
>
>
> R3: The section on 'Shaping the cost or reward' was not clear enough to me (please expand).
>
> >> We can improve the prose here. This section is pointing to works that acknowledge there may be weakness in the current setup of the optimization problem. To solve this limitation, they fine tune the cost  or reward function used in control, rather than trying to change the dynamics model.

---

### Decision · Program_Chairs · 2019-12-19

**Decision:**

Reject

**Comment:**

As the reviewers point out, this paper has potentially interesting ideas but it is in too preliminary state for publication at ICLR.